# Superior Efficacy of Apathogenic Genotype I (V4) over Lentogenic Genotype II (LaSota) Live Vaccines against Newcastle Disease Virus Genotype VII.1.1 in Pathogen-Associated Molecular Pattern-H9N2 Vaccinated Broiler Chickens

**DOI:** 10.3390/vaccines11111638

**Published:** 2023-10-25

**Authors:** Ahmed Elbestawy, Hany Ellakany, Mahmoud Sedeik, Ahmed Gado, Mervat Abdel-Latif, Ahmed Noreldin, Ahmed Orabi, Ismail Radwan, Wafaa Abd El-Ghany

**Affiliations:** 1Department of Poultry and Fish Diseases, Faculty of Veterinary Medicine, Damanhour University, Damanhour 22511, Egypt; ellakany_hany@vetmed.dmu.edu.eg (H.E.); ahmed.gado@vetmed.dmu.edu.eg (A.G.); 2Department of Poultry and Fish Diseases, Faculty of Veterinary Medicine, Alexandria University, Edfina 22758, Egypt; mahmoud.seddek@alexu.edu.eg; 3Nutrition and Veterinary Clinical Nutrition Department, Faculty of Veterinary Medicine, Damanhour University, El-Beheira 22511, Egypt; mervat.abdellatif@vetmed.dmu.edu.eg; 4Department of Histology and Cytology, Faculty of Veterinary Medicine, Damanhour University, El-Beheira 22511, Egypt; ahmed.elsayed@damanhour.edu.eg; 5Department of Microbiology, Faculty of Veterinary Medicine, Cairo University, Giza 12211, Egypt; orabivet@cu.edu.eg; 6Department of Bacteriology, Mycology and Immunology, Faculty of Veterinary Medicine, Beni-Suef University, Beni Suef 62511, Egypt; ismail.saad@vet.bsu.edu.eg; 7Department of Poultry Diseases, Faculty of Veterinary Medicine, Cairo University, Giza 12211, Egypt; wafaa.soliman@cu.edu.eg

**Keywords:** vNDV-VII.1.1 challenge, broiler chickens, live ND vaccines, V4, LaSota, PAMP-H9N2

## Abstract

A comparison of the efficacy of apathogenic genotype I (V4) and lentogenic genotype II (LaSota) strains of live Newcastle disease virus (NDV) vaccines was performed following vaccination with pathogen-associated molecular pattern (PAMP) H9N2 avian influenza vaccine and challenge with velogenic NDV genotype VII.1.1 (vNDV-VII.1.1). Eight groups (Gs) of day-old chicks were used (*n* = 25). Groups 1–4 received a single dose of PAMP-H9N2 subcutaneously, while Gs (1, 5) and (2, 6) received eye drops of V4 and LaSota, respectively, as two doses. All Gs, except for 4 and 8, were intramuscularly challenged with vNDV-VII.1.1 at 28 days of age. No signs were detected in Gs 1, 5, 4, and 8. The mortality rates were 0% in Gs 1, 4, 5, and 8; 40% in G2; 46.66% in G6; and 100% in Gs 3 and 7. Lesions were recorded as minimal in Gs 1 and 5, but mild to moderate in Gs 2 and 6. The lowest significant viral shedding was detected in Gs 1, 2, and 5. In conclusion, two successive vaccinations of broilers with a live V4 NDV vaccine provided higher protection against vNDV-VII.1.1 challenge than LaSota. PAMP-H9N2 with live NDV vaccines induced more protection than the live vaccine alone.

## 1. Introduction

Newcastle disease (ND) is an acute highly contagious viral disease of poultry. Over the last decade, ND has caused continuous devastating effects on the global poultry industry; therefore, the World Organization for Animal Health (WOAH) listed it as a notifiable terrestrial animal disease [1]. The incidence of ND has increased because of improper vaccination programs with vaccination failure, the presence of immunosuppressive diseases, and the genetic mutations of the virus, which lead to changes in biological characteristics and pathogenicity, even in vaccinated flocks [2].

Newcastle disease virus (NDV) or avian Orthoavulavirus 1 (AOAV-1) has been recently classified [3]. Based on the phylogenetic analysis of the fusion (F) protein gene sequence, NDV strains are divided into two major classes, namely class I and class II. Class I comprises non-virulent strains and consists of one genotype and three sub-genotypes [4]. However, class II includes virulent and non-virulent strains and consists of at least 20 genotypes (I–XXI) and multiple sub-genotypes following the exclusion of genotype XV, which contains recombinant sequences [5,6]. Genotype VII.1.1 of NDV is now regarded as the predominant circulating genotype in various poultry species in Egypt, as well as several African, Middle East, Asian, and European countries [7,8,9,10].

The outer surface proteins, such as hemagglutinin-neuraminidase (HN) and F, which are located on the NDV envelope, are important for viral virulence, tropism, and immunoprotection through neutralizing antibodies [11,12,13]. The HN protein is important for the initial viral binding to the sialic acid receptors of host cells, while F protein mediates the fusion of the viral envelope with the cell membrane [1].

Both vaccination and biosecurity measures are major parallel strategies for the prevention and control of NDV infections in endemic areas. Avirulent and lentogenic strains of NDV have been extensively used as live vaccines that can efficiently cross-protect against the velogenic and mesogenic pathotypes of the virus [14]. Although all NDV isolates are serologically grouped into a single serotype [1], the protection induced by the vaccine is not always optimal. The inadequate efficacy of NDV vaccines may be due to poor vaccination practices, improper biosecurity measures, or the antigenic differences (neutralizing epitopes) between the circulating virulent viruses and the vaccine strains [13,14,15,16]. Antigenic similarity between the circulating velogenic (vNDV) and the vaccine strain could improve the induced vaccinal protection in terms of reduced infectivity and virus shedding [6,17,18,19]. Vaccination with a live genotype I-based vaccine (ND.TR.IR strain) has protected broiler chickens against the local heterologous genotype VII vNDV and Herts 33 strain in Iran [20,21].

Chicken macrophages express several receptors for the recognition of pathogens, including Toll-like receptors (TLRs). These receptors can recognize or bind to pathogen-associated molecular patterns (PAMP) or synthetic ligands, leading to activation of macrophages, which play a critical role in immunity against viruses [22,23,24]. Previously, a whole inactivated virus (WIV) vaccine for H5N1, prepared by natural pathogen-associated molecular pattern (PAMP) technology, was more immunogenic and induced protective antibody responses at a lower antigenic dose than other formulations, like split virus or subunit vaccines, through the stimulation of Toll-like receptors (TLRs) of the innate immune system, in particular TLR7 [25].

To the best of our knowledge, scanty information is available about the comparative efficacy of genotypes I- and II-based live NDV vaccines in the protection of broiler chickens against vNDV-VII.1.1 challenge. Therefore, the objectives of this research were the investigation of the protective efficacy of V4 (an apathogenic genotype I-) and LaSota (a lentogenic genotype II-) based live NDV vaccines against vNDV-VII.1.1 challenge in broiler chickens that were previously vaccinated with inactivated low-pathogenic avian influenza virus subtype H9N2 prepared by PAMP technology (PAMP-H9N2).

## 2. Materials and Methods

### 2.1. Ethical Approval

All experimental procedures complied with the general guidelines of the Local Ethics Commission of the Institutional Animal Care and Use Committee of Faculty of Veterinary Medicine, Cairo University (Vet IA CUC) and the ethical approval code is Vetcu09092023742. All efforts have been made to minimize chicken suffering.

### 2.2. Challenge Virus

The challenge vNDV strain was isolated from a 26-day-old broiler chicken flock of 26,000 birds in El- Beheira governorate, Egypt. The flock was vaccinated against NDV using live vaccines only and exhibited a mortality rate of 10% within 5 days before sampling. The virus was identified as AOAV-1 (NDV) genotype VII.1.1 (AOAV-1/Egy/Ch/MR78/2018), and the full F protein gene sequencing revealed the velogenic motif ^112^RRQKRF^117^ (GenBank accession No. MK984238). The intracerebral pathogenicity index of the isolated vNDV in chickens was 1.86 and the mean death time in the embryonated chicken eggs (ECEs) was 40 h [26]. The challenge virus was propagated and titrated in 10-day-old specific pathogen-free ECEs [27], and the required challenge dose was then prepared according to WOAH [28].

### 2.3. Birds’ Management, Experimental Design, and Evaluation Parameters

A total of 200 one-day-old commercial broiler chicks (Ross 308) of mixed sex were divided into 8 equal groups (Gs), *n* = 25 each (Table 1). Each group was floor-reared in separate clean rooms. Groups 1–4 were subcutaneously vaccinated with an inactivated low-pathogenic AIV vaccine using PAMP technology (H9N2^®^P, MSD, Intervet Int., Noord-Brabant, The Netherlands, containing an inactivated avian influenza virus type A, subtype H9N2, strain A/CK/UAE/415/99; dose ≥ 8 Log_2_ HI) at the 4th day of age, while Gs 5–8 did not receive this vaccine. At the 8th and 15th days of age, chickens of Gs 1 and 5 were vaccinated with a live apathogenic V4 strain (Vaxsafe^®^ ND, Bioproperties, Glenorie, NSW, Australia; dose ≥ 10^6^ EID_50_), while those of Gs 2 and 6 were vaccinated with a live lentogenic LaSota strain (Avishield^®^ ND, Decra, London, UK; dose 10^6^–10^7^ TCID_50_) using the eye drop method for both vaccines. At 28 days old (DO), each bird in Gs 1, 2, 3, 5, 6, and 7 was intramuscularly challenged with 0.1 mL containing 10^6^ embryo lethal dose (ELD_50_)/bird of vNDV-VII.1.1. However, Gs 4 and 8 were kept as controls without NDV vaccination or challenge. During the experiment, the chickens were fed on a standard diet that met their nutritional requirements [29,30] and they had free access to fresh water ad libitum.

#### 2.3.1. Clinical Observation and Performance Parameters

The clinical signs, gross lesions, and mortality were observed daily for 10 days post-challenge (dpc). Additionally, the performance parameters included the weekly recorded body weights (Bwts), feed intake (FI), and the weekly calculated feed conversion ratio (FCR). Clinical disease scores were calculated for different groups in accordance with the intracerebral pathogenicity index score calculation [31].

#### 2.3.2. Histopathological Assessment

Two birds in each group were sacrificed humanely through cervical dislocation after the intravenous injection of sodium pentobarbital with a dose of 50 mg/kg at 0, 3, 7, 10 and 14 dpc; then, their tracheas, lungs, thymus glands, spleens, and bursa of Fabricius were collected, flushed with sterile phosphate-buffered saline (PBS, pH 7.4), and fixed in neutral buffered formol solution (10%) for 48 h at 4 °C. The fixed specimens were processed by the conventional paraffin embedding technique, including dehydration, through ascending grades of ethanol, clearing in three changes in xylene, and embedding in paraffin wax at 65 °C. Four-micrometer-thick sections were stained by Hematoxylin and Eosin (H and E) as previously described by Bancroft and Layton [32]. The microscopical examination was performed using a digital camera (Leica EC3, Leica, Wetzlar, Germany) connected to a microscope (Leica DM500, Deerfield, IL, USA) and software (Leica LAS EZ, 3.0). Semiquantitative lesion scoring of organs was calculated according to Gibson-Corley et al. [33]. Briefly, 10 fields were randomly chosen from each investigated group where the lesions were scored in a blind way by an expert (Score scale: 0 = normal; 1 ≤ 25%; 2 = 26–50%; 3 = 51–75%; 4 = 76–100%) and then averaged.

#### 2.3.3. Evaluation of the Virus Shedding

Individual tracheal (*n* = 3) and cloacal (*n* = 3) swabs were collected at 0, 3, 7, and 10 dpc for the detection of NDV shedding using real-time polymerase chain reaction (rRT-PCR). Briefly, each swab was immersed in a tube containing 2 mL of PBS (supplemented with antibiotic and antifungal agents) and shaken to obtain a suspension, which was stored at −80 °C until use. The viral RNA was extracted using a viral RNA kit QIAamp Viral RNA Mini Kit (Qiagen, Hilden, Germany) according to the manufacturer’s protocol. The used oligonucleotide primers and probes [34] were supplied by Metabion according to the manufacturer’s instructions. rRT-PCR was conducted using Strategen 3005P, Sacramento, CA, USA [8].

#### 2.3.4. Serological Evaluation

Eight blood samples were collected from the wing veins of chickens in each group on a weekly basis. Serum was separated to determine the antibody titers against NDV and low-pathogenic avian influenza subtype H9N2 (LPAI-H9N2) using the hemagglutination inhibition (HI) test [28,35]. Standard antigens for NDV genotype II and LPAI-H9N2 were used with hemagglutination titers of 8 log_2_. Four hemagglutination units were applied during the HI test.

#### 2.3.5. Immune Mediators Analysis

Five blood samples were collected from each chicken group at 17, 21, 28, and 31 days old and tested for the detection of some immune mediator levels using enzyme-linked immuno-sorbent assay (ELISA) kits based on the principle of double-antibody sandwich technology. These immune mediators included interleukin-1β (IL-1β) [36] (Novatein Bio, Woburn, MA, USA), chicken cluster of differentiation 4 (CD4) [37] (Shanghai Coon Koon Biotech. Co., Ltd., Shanghai, China), lysozyme (LYZ) [38] (Shanghai Coon Koon Biotech. Co., Ltd., Shanghai, China), and nitric oxide (NO) [39] (Shanghai Coon Koon Biotech. Co., Ltd., Shanghai, China). Absorption was measured at a wavelength of 450 nm. In addition, 2 spleen samples were collected from each chicken group at 31 days old and their homogenates were examined for the gene expression of cytokines IL-4, IL-10, and interferon-γ (INF-γ). Primers and cycling conditions were used [40,41,42].

#### 2.3.6. Statistical Analysis

All data calculations were statistically analyzed using the SPSS programming tool (IBM SPSS.20^®^) (SPSS Inc., Chicago, IL, USA) through one-way ANOVA followed by Duncan’s multiple range tests. GraphPad Prism 7^®^ was used to analyze the data in Figures 1, 7, and 8. All significant deviations were based on *p* ≤ 0.05 [43].

## 3. Results

### 3.1. Clinical Observation and Performance Parameters

The typical clinical picture of NDV, such as sneezing, rales, nasal and ocular discharge, conjunctivitis, coughing, head swelling, greenish diarrhea, torticollis, and lateral recumbency, in addition to tracheitis, pneumonia, proventricular hemorrhages, enteritis, petechial hemorrhages on ileocecal tonsils, splenitis, hepatic congestion with distended gall bladder, and nephritis, were observed from the 2nd dpc and 5th dpc (nervous signs) in G3 (PAMP-challenged) and G7 (non-vaccinated-challenged), respectively. However, fewer clinical signs were observed in G2 (PAMP-LaSota-challenged) and G6 (LaSota-challenged) than in the control positive Gs 3 and 7. It is important to note that chickens of G1 (PAMP-V4-challenged) and G5 (V4-challenged) showed only a slight decrease in feed intake. The significant minimum (*p* ≤ 0.05) clinical disease scores were recorded in Gs 4 and 8 as 0 and in Gs 1 and 5 as 0.1, followed by consecutively higher scores (*p* ≤ 0.05) of 0.6 in G2, 0.8 in G6, 1.5 and 1.6 in Gs 3 and 7, respectively (Appendix A). Regarding mortality rates, they were 0% in Gs 1, 4, 5, and 8; 40% in G2; 46.66% in G6, and 100% for both Gs 3 and 7 during 7 dpc (Figure 1 and Appendix A). The highest average final Bwts (*p* ≤ 0.05) were recorded in G4 (1985a g), G8 (1980a g), G5 (1968a g), and G1 (1944a g) compared with G6 (1496b g) and G2 (1465b g). Moreover, the FCR was recorded as 1.52, 1.55, 1.57, 1.58, 1.79, and 1.86 for Gs 1, 5, 4, 8, 2, and 6, respectively.

### 3.2. Histopathology

Tracheal epithelium and cilia were normal and well-developed in the chickens of control negative Gs 8 and 4 from 3 to 14 dpc (Figure 2A,B). On the other hand, the challenged chickens of G7 showed severe necrotic tracheitis, desquamation of hyperplastic hemorrhagic tracheal mucosa, and lymphocytic infiltration from 3 to 14 dpc (Figure 2C). Likewise, the challenged chickens of G3 exhibited severe tracheal hyperplasia with a complete loss of the cilia and metaplasia of surface epithelia into squamous cells from 3 to 14 dpc (Figure 2D). Likewise, the LaSota vaccinated-challenged chickens of G6 displayed diffuse infiltration by lymphocytes, loss of cilia, and severe hemorrhage and degeneration of respiratory epithelia from 3 to 7 dpc and mild pathologic lesions on 10 and 14 dpc (Figure 2E). H9N2-PAMP and LaSota vaccinated-challenged chickens of G2 showed fewer lesions in terms of slight hyperplasia and metaplasia of tracheal mucosa with slight degeneration of cilia from 3 to 7 dpc and mild pathologic lesions on 10 and 14 dpc (Figure 2F). In G5 (V4 vaccinated-challenged), the chickens exhibited relatively normal tracheal epithelia with slight lymphocytic infiltration and normal cilia from 3 to 10 dpc (Figure 2G). Moreover, complete protection of trachea in the form of normal epithelia was displayed in chickens of G1 (H9N2-PAMP and V4 vaccinated-challenged) from 3 to 14 dpc (Figure 2H).

The histopathological examination of the lungs of chickens in Gs 4 and 8 revealed normal alveoli without any pathological lesions from 3 to 14 dpc (Figure 3A,B). However, chickens in G7 had severe hemorrhages, interstitial fibrous pneumonia, and extensive lymphocytic infiltration from 3 to 14 dpc (Figure 3C). Vaccination with H9N2-PAMP in G3 slightly decreased the severity of lesions with no fibrous pneumonia, but severe hemorrhage and lymphocytic infiltration were present from 3 to 14 dpc (Figure 3D). In contrast, pre-vaccination with LaSota either with (G2) or without (G6) H9N2-PAMP simply protected against vNDV-VII.1.1 challenge with moderately congested blood vessels from 3 to 7 dpc and mild pathologic lesions on 10 and 14 dpc (Figure 3E,F). Furthermore, relatively normal lungs were detected in V4-vaccinated G1 or G5 from 3 to 14 dpc (Figure 3G,H). 

The thymic architecture of the cortex and medulla in the chickens of Gs 4 and 8 was normal, without any detected lesions, from 3 to 14 dpc (Figure 4A,B). In contrast, vNDV-VII.1.1-infected chickens of G7 displayed destructed thymic architecture in the form of lymphoid depletion, necrosis in the whole lobule with macrophage accumulation, and severe congestion from 3 to 14 dpc (Figure 4C). The thymic architecture was slightly protected in H9N2-PAMP-vaccinated chickens of G3, but hemorrhages were still found and several Hassell’s corpuscles were extended to the thymic cortex from 3 to 14 dpc (Figure 4D). In LaSota-vaccinated Gs 2 and 6, the thymic architecture was more organized into the cortex and medulla, with several congested blood vessels and many Hassell’s corpuscles from 3 to 14 dpc (Figure 4E,F). The best protection of thymic architecture was seen in both V4-vaccinated chickens of Gs 1 and 5 from 3 to 14 dpc (Figure 4G,H).

Chickens of control negative Gs 4 and 8 did not have any lesions in the spleen from 3 to 14 dpc (Figure 5A,B). On the other hand, severe multifocal lymphoid depletion and necrosis were detected in challenged chickens of both Gs 3 and 7 from 3 to 14 dpc (Figure 5C,D). The vaccination with LaSota did not protect against vNDV-VII.1.1 challenge in G6 from 3 to 14 dpc (Figure 5E), but H9N2-PAMP and LaSota vaccinations induced mild necrosis and lymphoid depletion in G2 from 3 to 14 dpc (Figure 5F). The splenic architecture was normal after challenge and V4 vaccination of Gs 1 and 5 from 3 to 14 dpc (Figure 5G,H).

The bursa of Fabricius was normal in chickens of both control negative Gs 4 and 8 from 3 to 14 dpc (Figure 6A,B). However, vNDV-VII.1.1 infection in G7 caused complete damage to the bursal architecture in terms of severe lymphoid depletion, necrosis, infiltration of inflammatory cells, and complete destruction of the covered epithelium from 3 to 14 dpc (Figure 6C). In G3, moderate lymphoid depletion, necrosis, and thickening of the interfollicular septum with congested blood vessels were prominent from 3 to 14 dpc (Figure 6D). LaSota vaccination in G6 lowered lesions caused by vNDV that were represented by mild lymphoid depletion and necrosis from 3 to 14 dpc (Figure 6E). However, chickens of G2 were more protected against challenge from 3 to 14 dpc (Figure 6F). The bursal architecture in V4-vaccinated Gs 1 and 5 was normal, as the negative control groups, from 3 to 14 dpc (Figure 6G,H). More details for all examined organs are provided in a low power (Appendix A). The histopathological lesion scores were significantly decreased (*p* ≤ 0.05) in chickens of Gs 1 and 5, respectively, compared with all other vNDV-VII.1.1-challenged chickens. It was noticed that PAMP-H9N2-vaccinated chickens had lower lesion scores compared with non-vaccinated ones, as indicated in Table 2.

### 3.3. Virus Shedding

The lowest tracheal shedding (*p* ≤ 0.05) was detected at the 3rd dpc in chickens of G1 (1.3 log_10_ viral copies), followed by G2 and G5 (2 log_10_), compared with G3, G6 (3.9 log_10_), and G7 (3.2 log_10_). The tracheal shedding continued until the 7th dpc in Gs 2, 5, and 6. Moreover, cloacal shedding decreased significantly in Gs 5 (0.7 log_10_), 1 (1.6 log_10_), and 2 (2 log_10_) compared with Gs 3, 6 (3 log_10_), and 7 (2.7 log_10_) at the 3rd dpc. Chickens in both Gs 2 and 6 showed cloacal shedding until the 7th and the 10th dpc (Figure 7).

### 3.4. Serology

At 28 DO, the highest HI titers (*p* ≤ 0.05) for NDV were found in chickens of Gs 5, 1, and 2 as 5.25, 4.75, and 4.13 log_2_, respectively. However, at 35 DO (7th dpc), the titers increased (*p* ≤ 0.05) to 11, 10.63, 9.38, and 8 log_2_ in Gs 6, 2, 5, and 1, respectively, compared with the other groups. In addition, at 35 DO, the highest HI titers (*p* ≤ 0.05) for LPAIV (H9N2) were detected in Gs 1, 4, and 2 as 7.25, 6.88, and 6 log_2_, respectively, compared with the other groups (Table 3).

### 3.5. Immune Mediators

The results of the IL-1β, CD4, LYZ, and NO levels at 17, 21, and 28 DO indicated a higher significant difference (*p* ≤ 0.05) in the level of IL-1β in PAMP-vaccinated Gs 1, 4, and 3 compared with the non-PAMP-vaccinated Gs 5–8. However, on the 3rd dpc, the results of IL-1β did not differ significantly (*p* > 0.05) in all groups, but they were numerically higher in G2 (319.6 pg/mL) and G3 (318.4 pg/mL). At 31 DO, the highest difference (*p* ≤ 0.05) in the CD4 level was observed in G8 (7.2 ng/mL), followed by Gs 5 and 7 (6 ng/mL), Gs 1 and 2 (5.6 ng/mL), as well as Gs 3 and 6 (5 ng/mL), respectively. For LYZ and NO expressions, G7 had the highest (*p* ≤ 0.05) levels (132.6 ng/mL and 82.6 μmol/L, respectively) on the 3rd dpc (Table 4).

The gene expression of IL-4 and IL-10 in spleen showed a higher significant difference (*p* ≤ 0.05) in Gs 1–6 in comparison with controls Gs 7 and 8. For INF-γ, the lowest gene expression was recorded in G8 (Ct value of 0), while the highest significant (*p* ≤ 0.05) expression was recorded in G7 (the lowest Ct value of 5.6); followed by Gs 1, 2, and 6 (Ct values of 14, 15, and 15.75, respectively), and then Gs 5 and 3 (Ct values of 18 and 24.5, respectively). However, the lowest significant (*p* ≤ 0.05) INF-γ gene expression (the highest significant (*p* ≤ 0.05) Ct value of 27.5) was reported in G4 (Figure 8).

## 4. Discussion

Despite the adoption of various vaccines and extensive vaccination programs for controlling ND, it still causes high and worldwide economic losses in the poultry industry. The low-virulence strains of live NDV vaccines, such as LaSota and B1 (pneumotropic), and V4 and VG/GA (enterotropic), have been commonly used to control virus infections. Generally, the pneumotropic NDV vaccines in chickens are more virulent and usually tend to be more immunogenic than the enterotropic types [14,44,45]. Although NDV is one serotype, several genotypes have been recorded. NDV strains are divided into two major classes, namely class I and class II. Class I comprises non-virulent strains and consists of one genotype and three sub-genotypes. However, class II includes virulent and non-virulent strains and consists of at least 20 genotypes (I–XXI) and multiple sub-genotypes following the exclusion of genotype XV, which contains recombinant sequences [4]. The outer surface proteins, such as hemagglutinin-neuraminidase (HN) and F, which are located on the NDV envelope, are important for viral virulence, tropism, and immune protection through neutralizing antibodies. The antigenic differences in neutralizing epitopes between the circulating virulent viruses (e.g., genotype VII.1.1) and the vaccine strains may be responsible for the inadequate vaccine efficacy [4,13,15,16]. Furthermore, the higher matching percent of aa in the F and HN proteins of NDV vaccines and the field virus is essential for enhancing vaccine protection in terms of reduced infectivity and minimized viral shedding rate [6,14,15,17,18,19,46,47,48]. Previously, high protection and a reduction in viral shedding have been reported in birds vaccinated with recombinant vaccines containing genotype I (D26) and challenged by heterologous genotypes VII.1.1, VII.2, and V of NDV [49,50,51,52]. Therefore, in order to confirm the theory of higher genetic and antigenic matching within F and HN proteins, which is essential for higher protection, this study was planned to evaluate the protective efficacy induced by genotype I (V4) or genotype II (LaSota) strains of live NDV vaccines against vNDV genotype VII.1.1 challenge in broiler chickens. In addition, the efficacy of using a combined vaccination with PAMP-H9N2 and live NDV vaccines was demonstrated here.

During 10 dpc, genotype I (V4) vaccinated-challenged Gs 1 and 5 were completely protected against NDV-induced clinical disease (the lowest significant (*p* ≤ 0.05) clinical disease scores) and mortalities, but genotype II (LaSota) vaccinated-challenged Gs 2 and 6 showed some signs and lesions and had higher clinical disease scores (*p* ≤ 0.05), in addition to mortality rates of 40% and 46.66%, respectively, despite the presence of 4.13 log_2_ mean HI titers in G2 at the time of challenge (28 days old). However, non-vaccinated and challenged chickens (Gs 3 and 7) displayed severe signs (the highest clinical disease scores (*p* ≤ 0.05)), with a mortality rate of 100%. The study by Sultan et al. [53] showed a mortality rate of 23% in broiler chickens following intranasal challenge by vND genotype VII.1.1 at 28 days old and vaccination with both live LaSota and inactivated genotype II NDV vaccines, although the mean HI titer at the time of challenge was 4.2 log_2_. The higher mortality rates in LaSota-vaccinated chickens in our study might be attributed to the double vaccinations with LaSota live vaccine only without any inactivated vaccine and the intramuscular route of challenge vs. the intranasal route used by Sultan et al. The absence of or significant reduction in clinical picture in vaccinated chickens Gs 1, 5, 2, and 6 was positively reflected in the Bwts and FCR. Histopathologically, chickens of V4-vaccinated Gs 1 and 5 showed the best protection against the virus challenge, with minimal pathological lesions and nearly normal architectures of all organs during all days post-challenge. However, some lesions, including apoptosis, hemorrhages in the lung, spleen, and germinal center of the bursa of Fabricius, degenerated thymus glands, and detached tracheal epithelium have been detected in chickens of LaSota-vaccinated Gs 2 and 6 compared with the challenged control Gs 3 and 7, which displayed more excessive lesions at the 3rd and 7th dpc. Some previous Egyptian reports confirmed decreases in lesion scores with a high degree of protection in chickens vaccinated with different genotypes matching vaccines and challenged by vND-VII.1.1 [23,53,54,55]. Moreover, PAMP vaccination minimized the histopathological lesions in chickens of G2 (LaSota vaccinated) compared with its parallel non-PAMP-vaccinated G6. This could be attributed to the enhancement in innate and adaptive immunity induced by PAMP vaccination, which was also reflected in the serological response at time of challenge, 28 days old (4.13 vs. 2.13 log_2_ in G2 and 6, respectively). It is important to link the results of histopathology, virus shedding, serological immune response, and cytokines production in different groups. A complete reduction in virus shedding was a main indication of the vaccine efficacy during the disease control in houses rearing chickens [51]. Here, it was recorded that V4-vaccinated G1 chickens had the lowest and shortest tracheal and cloacal shedding. Moreover, chickens vaccinated with both PAMP-H9N2 and live NDV vaccines had shorter viral shedding compared with those receiving live NDV vaccine alone. In the same way, Bello et al. [56] and Mohamed et al. [57] reported that both the duration and load of cloacal and oropharyngeal shedding of vND genotype VII virus challenge were significantly reduced when the genotype-matched vaccine was applied. Regarding the results of HI titers, the highest neutralizing antibodies (*p* ≤ 0.05) against the challenge virus were found in V4-vaccinated G1 (HI titer of 8 Log_2_) and G5 (HI titer of 9.38 Log_2_), rather than LaSota-vaccinated G2 (HI titer of 10.63 Log_2_) and G6 (HI titer of 11 Log_2_). This result indicates the lower efficacy of the LaSota vaccine in inducing high matched antibodies to neutralize the challenge virus. It has been reported that the neutralization titers against NDV are always 3- to 6-fold higher against a homologous virus than a heterologous one; additionally, the cross-reactivity R-value between aSG10 (genotype VII NDV) and LaSota strain was 0.23. Despite both aSG10 and LaSota strains having the same single serotype, they are loosely related, with major antigenic differences [18]. Moreover, the vaccination–challenge study revealed that the aSG10 vaccine strain showed a significantly higher protection rate than the LaSota vaccine strain when chickens were challenged with a genotype VII-NDV using the intramuscular or eye drop/intranasal route [18]. Further, Shahar et al. [19] proved that the differences between the NDV vaccine and the field viruses in the neutralizing epitopes (HN and F proteins) may enable the escape of the field variants strains from the vaccine antibodies. Based on the viral sequences, NDV isolates may be divided into “neutralization types” or “neutrotypes”, which can aid in the development of mutation-adjusted vaccines [19]. From our point of view, the effective viral neutralization in both Gs 1 and 5 led to low stimulation of IL-1β, mild pro-inflammation, and consequent mild stimulation of the antigen-presenting cells. Such mild stimulation decreased the release of LYZ and NO in comparison with other groups, particularly G7 and LaSota-vaccinated Gs 2 and 6. The critical innate and adaptive immunity of cytokines expression indicated an enhancement (*p* ≤ 0.05) in the IL-4 and IL-10 levels in the spleen of PAMP-vaccinated chickens (Gs 1–4) and live NDV-vaccinated chickens (Gs 5 and 6). High INF-γ gene expression (*p* ≤ 0.05) was recorded in chickens of Gs 1, 2, 5, 3, and 4, which indicates the anti-inflammatory effect. Previous research mentioned that the INF-γ of chickens plays an important role in the immunogenicity and pathobiology of NDV [58,59]. The superior protection of V4 could be also attributed to the higher local immunity induced in the intestinal tract (enterotropic strain) than that induced by LaSota (pneumotropic strain) against velogenic viscerotropic genotype VII.1.1 NDV. Positive correlations between the ND live vaccine distribution and its specific immunoglobulin A (IgA) production were observed. Perozo et al. [60] recorded higher IgA levels in the intestinal tract, bile, and intestine for the VG/GA (enterotropic, pathogenic) strain than for LaSota (pneumotropic, lentogenic), which induced higher levels in the trachea. The replication pattern of the VG/GA strain induced a stronger localized mucosal immune response, as shown by an increased production of NDV-specific IgA. This feature may represent a competitive advantage after challenge with velogenic viscerotropic NDV owing to the massive destruction of intestinal lymphoid areas and extensive ulceration of overlying intestinal epithelium associated with the active replication of this virus [61]. However, this point should be briefly illustrated in further research work. The higher gene expression and better histopathology were also reflected in the viral shedding, as G1 had the lowest and shortest tracheal and cloacal shedding. The main question of this study is how could the V4 (genotype-I) vaccine of NDV induce higher protection in chickens than LaSota. To answer this question, the role of both F and HN proteins in the induction of neutralizing antibodies against vNDV should be considered. The HN protein prevents the viral attachment step (anti-HN antibodies), while the F protein inhibits virus spread to nearby cells (anti-F antibodies), even when the virus enters the cell and multiplies within it [62,63,64]. Further, Kim et al. [12] reported that the F protein plays the most important role in the protection of virus infection following the use of genotype-matched vaccines. Tabatabaeizadeh [65] recorded that, the number of identical linear and conformational neutralizing epitopes of F and HN proteins (n = 15) of the vaccine strains compared with the VII.2 NDV were as follows: (V4 = 14) > (I-2 = 11) > (Ulster, VG/GA, and R2B = 10) > (F = 9) > (LaSota = 7) > (PHY.LMV.42 = 6) > (B1 = 5). In the same previous study, V4 and I-2 vaccine strains showed the highest number of identical epitopes compared with the other vaccine strains, while the B1, PHY.LMV.42, and LaSota strains revealed the lowest epitope identity compared with VII.2 NDV, “especially in F protein”. Similar results of aa differences between LaSota and VG/GA were recorded previously by [60]. Under immunological pressure, the estimation of evolutionary distance confirmed that the number of amino acid substitutions per site between sequences was higher and changed more rapidly for HN compared with F protein [66,67,68,69,70,71,72,73]. Additionally, the decrease in the identity of common epitopes between the vaccine and the vNDV field strains may result in expanded and non-matched protective antibodies, lower levels of virus neutralization, higher rate of viremia, higher severity of the disease, and prolonged virus load in organs [6,15,17,18,19]. In the recent study by Liu et al. [73], the results indicated that the amino acids similarity of the LaSota strain with vNDV genotype VII strain was 88.4% for the F protein and 88.5% for the HN protein, and that with vNDV genotype IX strain was 92.2% for the F protein and 91.1% for the HN protein. Moreover, the genotype IX strain is genetically closer to the LaSota strain than the genotype VII strain and, consequently, this higher antigenic similarity can provide better protection against the development of clinical disease and viral shedding.

Finally, it is essential to note that the adoption of strict biosecurity measures before the completion of the vaccine-derived immunity, control of the mixed viral/bacterial respiratory diseases and immunosuppressive agents, minimizing stressors, as well as application of efficient vaccines are crucial factors to prevent the infection with circulating NDV in poultry flocks [16].

## 5. Conclusions

Two successive vaccinations of commercial broiler chickens with apathogenic (genotype I-based, V4) live NDV vaccine fully protected chickens from the development of the clinical disease, mortalities, and histopathological lesions, lowered and shortened the viral shedding, and enhanced antibodies and cytokines production compared with a lentogenic (genotype II-based, LaSota) live vaccine. Chickens that received both PAMP-H9N2 and live NDV vaccines exhibited milder histopathological lesions and shorter viral shedding rate than those that received a live NDV alone.

## Figures and Tables

**Figure 1 vaccines-11-01638-f001:**
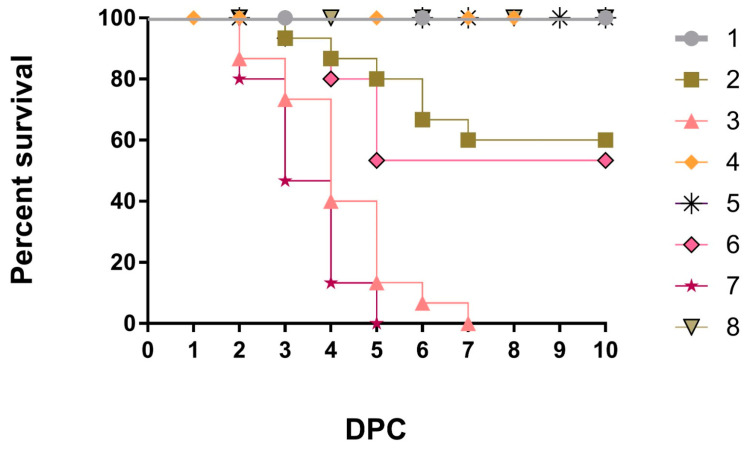
Survival rate in all chicken groups.

**Figure 2 vaccines-11-01638-f002:**
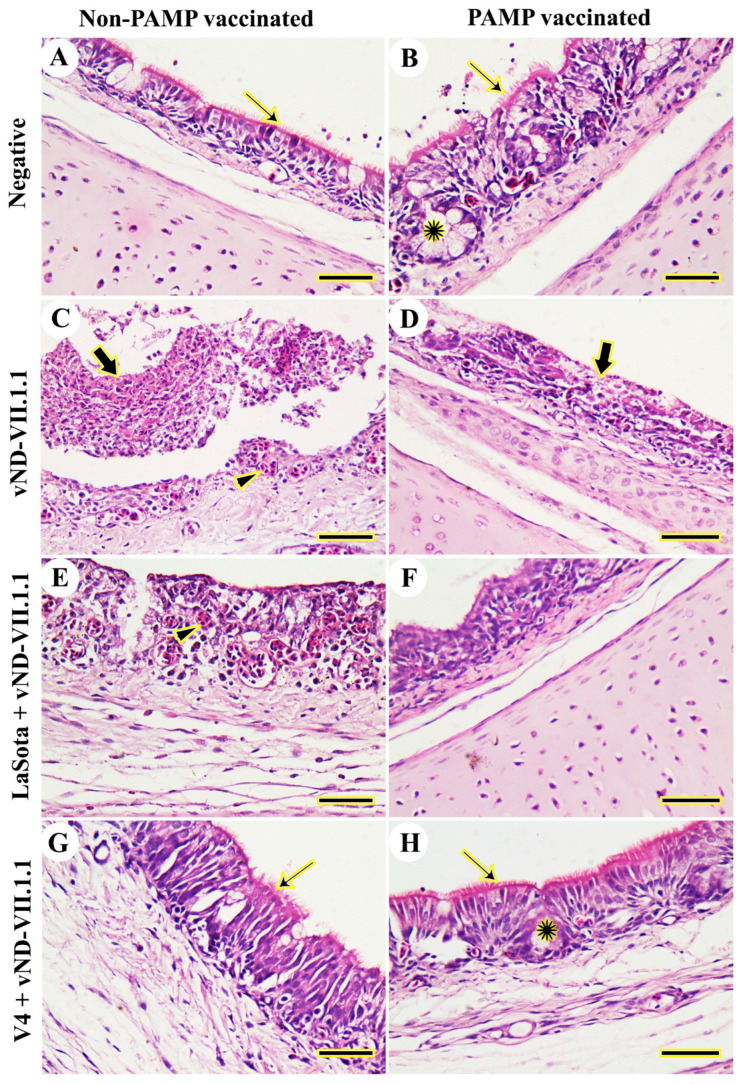
Histopathological examination of chicken tracheas. (**A**) Negative control (G8). (**B**) H9N2-PAMP vaccination only (G4). (**C**) vND-VII.1.1 challenge only (G7). (**D**) H9N2-PAMP vaccination and vND-VII.1.1 challenge (G3). (**E**) LaSota vaccination and vND-VII.1.1 challenge (G6). (**F**) H9N2-PAMP, LaSota vaccination, and vND-VII.1.1 challenge (G2). (**G**) V4 vaccination and vND-VII.1.1 challenge (G5). (**H**) H9N2-PAMP, V4 vaccination, and vND-VII.1.1 challenge (G1). Normal cilia (thin arrows), hemorrhage (arrowheads), normal tracheal glands (stars), and mucosal hyperplasia and metaplasia (thick arrows). Scale bar = 50 μm.

**Figure 3 vaccines-11-01638-f003:**
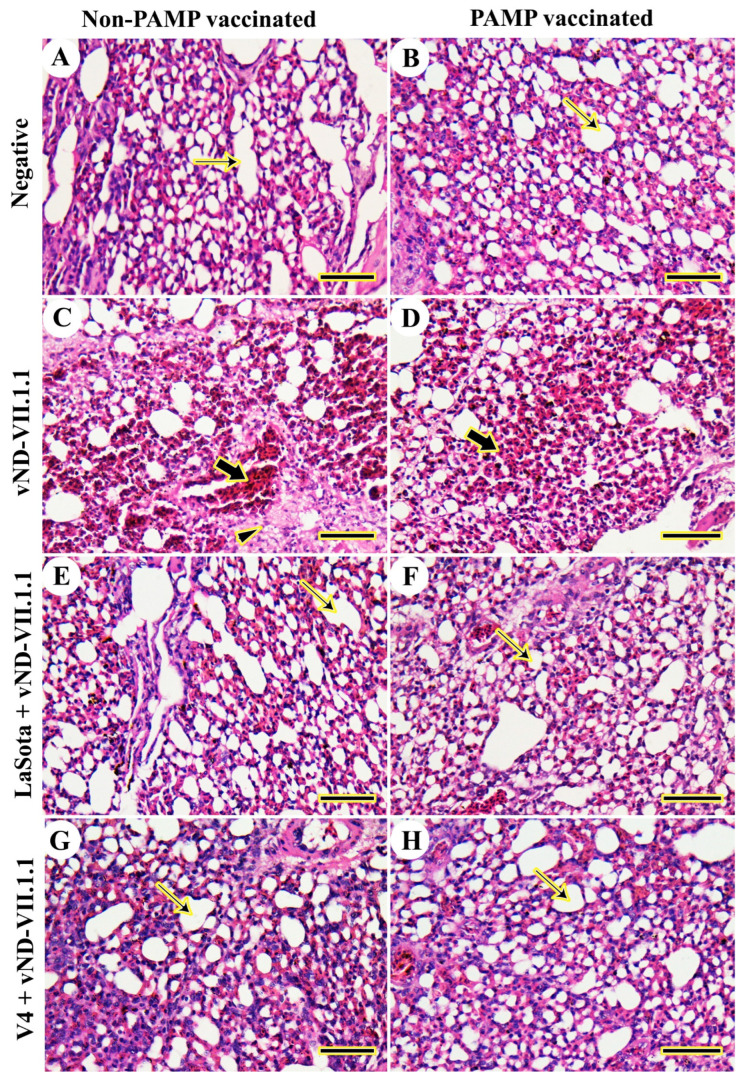
Histopathological examination of chicken lungs. (**A**) G8. (**B**) G4. (**C**) G7. (**D**) G3. (**E**) G6. (**F**) G2. (**G**) G5. (**H**) G1. Normal alveoli (thin arrows), thick interstitial connective tissue filled with inflammatory cells (arrowheads), and hemorrhage (thick arrows). Scale bar = 50 μm.

**Figure 4 vaccines-11-01638-f004:**
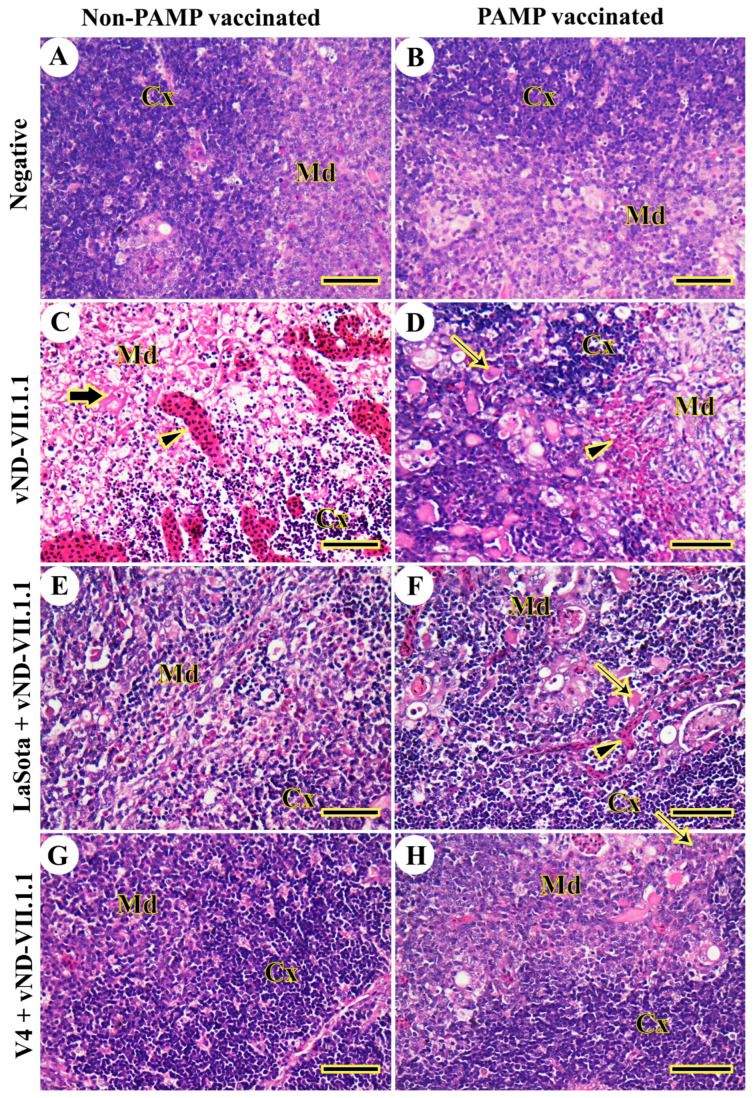
Histopathological examination of chicken’s thymus glands. (**A**) G8. (**B**) G4. (**C**) G7. (**D**) G3. (**E**) G6. (**F**) G2. (**G**) G5. (**H**) G1. Cortex (Cx), medulla (Md), Hassell’s corpuscles (thin arrows), severe congestion (arrowheads), and necrosis (thick arrows). Scale bar = 50 μm.

**Figure 5 vaccines-11-01638-f005:**
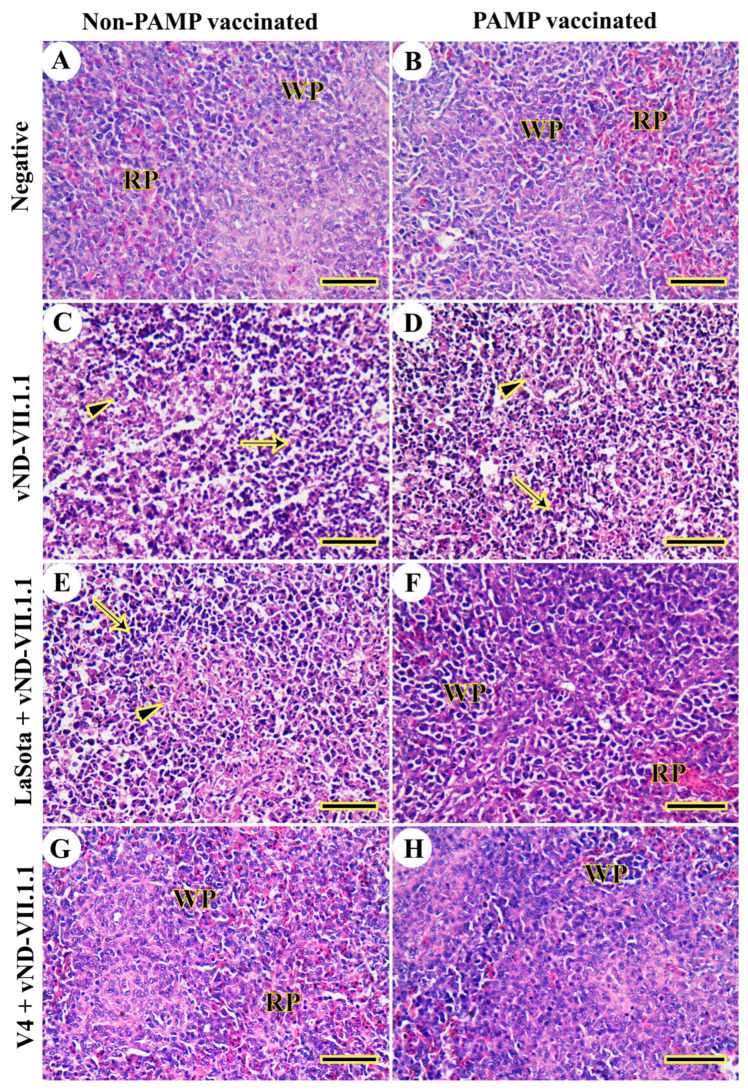
Histopathological examination of chicken spleens. (**A**) G8. (**B**) G4. (**C**) G7. (**D**) G3. (**E**) G6. (**F**) G2. (**G**) G5. (**H**) G1. Red pulp (Rp), white pulp (Wp), necrosis (thin arrows), and multifocal lymphoid depletion (arrowheads). Scale bar = 50 μm.

**Figure 6 vaccines-11-01638-f006:**
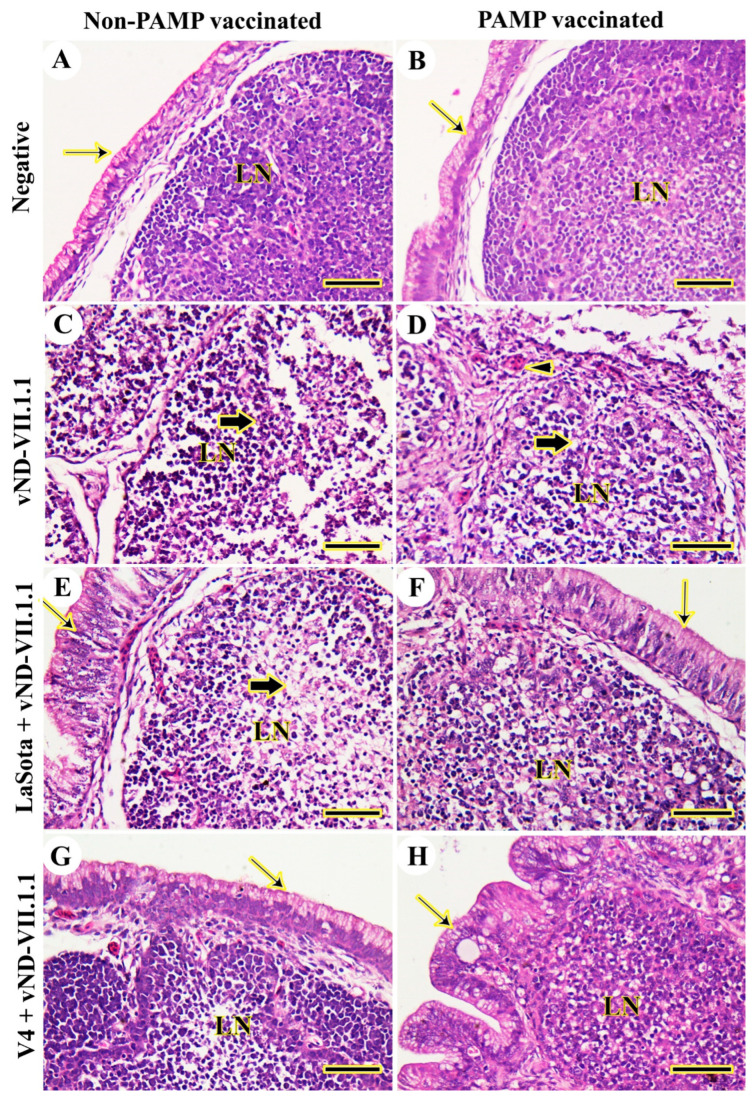
Histopathological examination of chicken bursa of Fabricius. (**A**) G8. (**B**) G4. (**C**) G7. (**D**) G3. (**E**) G6. (**F**) G2. (**G**) G5. (**H**) G1. Lymphoid nodule (LN), normal epithelium (thin arrows), necrosis and lymphoid depletion (thick arrows), and congested blood vessels (arrowheads). Scale bar = 50 μm.

**Figure 7 vaccines-11-01638-f007:**
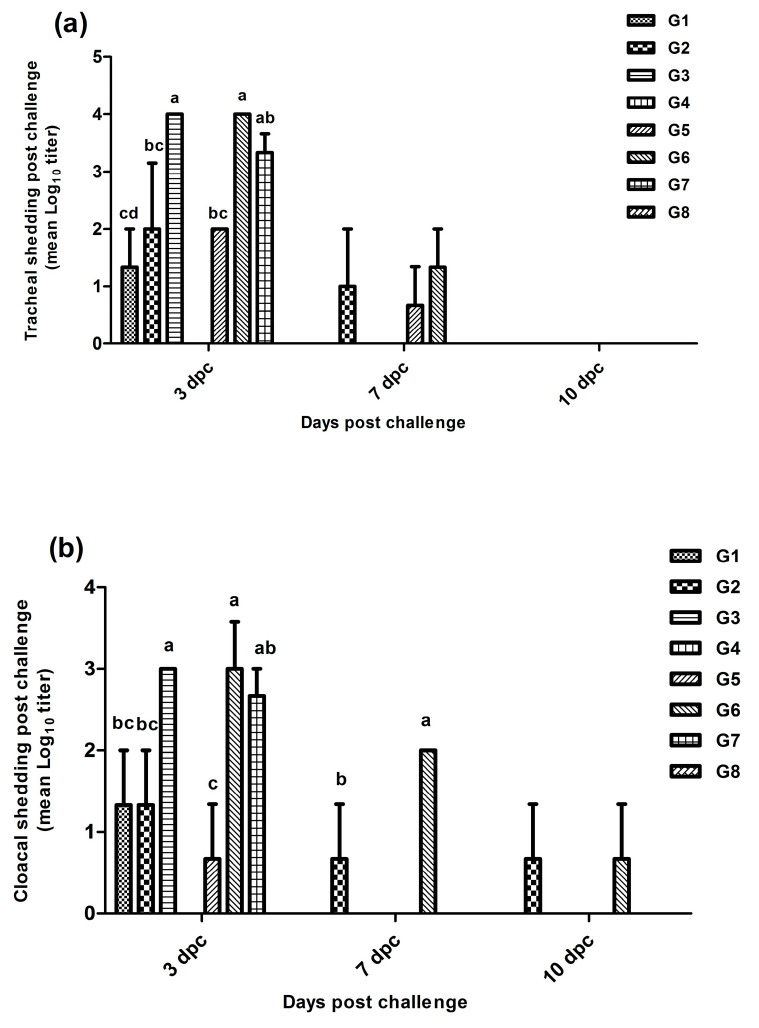
Tracheal (**a**) and cloacal (**b**) viral shedding in all groups at 3, 7, and 10 dpc.

**Figure 8 vaccines-11-01638-f008:**
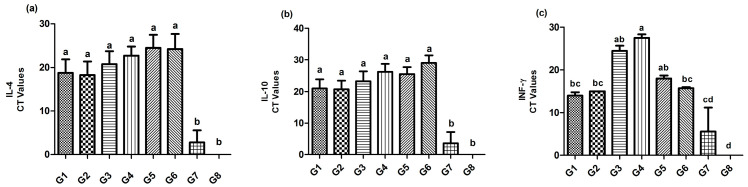
The detection of the IL-4 (**a**), IL-10 (**b**), and INF-γ (**c**) gene expression levels in the spleens of chickens at the 3rd dpc (31 DO) using rRT-PCR.

**Table 1 vaccines-11-01638-t001:** Experimental design.

Chicken Groups (Number)		Age (Days)			
		Vaccination			Challenge
		4	8	15	28
G1	25	H9N2-PAMP	V4	V4	vND-VII.1.1
G2	25	H9N2-PAMP	LaSota	LaSota	vND-VII.1.1
G3	25	H9N2-PAMP	-	-	vND-VII.1.1
G4	25	H9N2-PAMP	-	-	-
G5	25	-	V4	V4	vND-VII.1.1
G6	25	-	LaSota	LaSota	vND-VII.1.1
G7	25	-	-	-	vND-VII.1.1
G8	25	-	-	-	-

-: Not applied.

**Table 2 vaccines-11-01638-t002:** Histopathological lesion scoring of tracheae, lung, thymus glands, spleen, and bursa of Fabricius in all chicken groups.

Groups	Organ Histopathological Lesion Score						
	Trachea		Lung	Thymus Glands		Spleen	Bursa of Fabricius
	Hemorrhage	Mucosal Hyperplasia and Metaplasia	Hemorrhage	Congestion	Necrosis and Lymphoid Depletion	Necrosis and Multifocal Lymphoid Depletion	Necrosis and Lymphoid Depletion
1	0.3 ± 0.2 ^de^	0.5 ± 0.2 ^d^	0.30 ± 0.2 ^de^	0.6 ± 0.2 ^c^	0.3 ± 0.2 ^de^	0.6 ± 0.2 ^c^	0.3 ± 0.2 ^de^
2	2.1 ± 0.2 ^c^	2.5 ± 0.3 ^c^	2.0 ± 0.3 ^c^	3.1 ± 0.2 ^ab^	1.8 ± 0.2 ^c^	2.2 ± 0.2 ^b^	1.7 ± 0.2 ^bc^
3	2.8 ± 0.2 ^bc^	3.5 ± 0.2 ^ab^	3.4 ± 0.2 ^ab^	2.6 ± 0.2 ^b^	2.5 ± 0.2 ^bc^	3.2 ± 0.3 ^a^	3.3 ± 0.2 ^a^
4	0.0 ± 0 ^e^	0.0 ± 0 ^d^	0.0 ± 0 ^e^	0.0 ± 0 ^c^	0.0 ± 0 ^e^	0.0 ± 0 ^c^	0.0 ± 0 ^e^
5	1.0 ± 0.1 ^de^	0.6 ± 0.2 ^d^	0.9 ± 0.2 ^d^	0.8 ± 0.2 ^c^	1.0 ± 0.0 ^d^	0.7 ± 0.3 ^c^	1.0 ± 0.3 ^cd^
6	3.0 ± 0.3 ^ab^	2.9 ± 0.2 ^bc^	2.8 ± 0.3 ^b^	2.9 ± 0.2 ^ab^	3.2 ± 0.2 ^ab^	3.0 ± 0.2 ^ab^	2.4 ± 0.2 ^b^
7	3.8 ± 0.1 ^ab^	3.9 ± 0.1 ^a^	3.7 ± 0.2 ^a^	3.5 ± 0.2 ^a^	3.7 ± 0.2 ^a^	3.5 ± 0.2 ^a^	3.8 ± 0.2 ^a^
8	0.0 ± 0 ^e^	0.0 ± 0 ^d^	0.0 ± 0 ^e^	0.0 ± 0 ^c^	0.0 ± 0 ^e^	0.0 ± 0 ^c^	0.0 ± 0 ^e^

Means within each column for each division with common superscript letters are significantly different (*p* ≤ 0.05).

**Table 3 vaccines-11-01638-t003:** Serological response through HI titers for NDV and LPAIV-H9N2 in all chicken groups.

Groups	HI Titers for ND					HI Titers for LPAIV-H9N2				
	Age (Days)					Age (Days)				
	7	14	21	28	35	7	14	21	28	35
1	7.38 ± 1 ^a^	3.13 ± 0.4 ^bcd^	4.63 ± 0.4 ^a^	4.75 ± 0.4 ^a^	8.00 ± 0.3 ^c^	7.00 ± 0 ^b^	4.00 ± 0.3 ^ab^	2.13 ± 0.3 ^ab^	3.25 ± 0.9 ^ab^	7.25 ± 0.6 ^a^
2	6.88 ± 0.9 ^ab^	3.00 ± 0 ^cd^	2.13 ± 0.2 ^bc^	4.13 ± 0.5 ^a^	10.63 ± 0.3 ^ab^	11.13 ± 0.5 ^a^	4.00 ± 0.2 ^ab^	1.63 ± 0.3 ^abc^	1.63 ± 0.5 ^bcd^	6.00 ± 0.5 ^a^
3	6.50 ± 0.8 ^ab^	3.75 ± 0.3 ^abc^	1.75 ± 0.2 ^c^	1.00 ± 0.0 ^b^	0.00 ± 0 ^d^ (NA)	6.75 ± 0.4 ^b^	4.13 ± 0.1 ^a^	1.75 ± 0.2 ^bc^	5.00 ± 0.5 ^a^	0.00 ± 0 ^b^ (NA)
4	4.50 ± 0.2 ^b^	3.13 ± 0.1 ^bcd^	2.63 ± 0.7 ^bc^	1.38 ± 0.2 ^b^	0.13 ± 0.2 ^d^	7.63 ± 0.6 ^b^	3.75 ± 0.2 ^ab^	1.00 ± 0 ^c^	2.88 ± 0.4 ^bc^	6.88 ± 0.8 ^a^
5	4.63 ± 0.4 ^ab^	3.00 ± 0.0 ^cd^	4.25 ± 0.2 ^a^	5.25 ± 0.3 ^a^	9.38 ± 0.6 ^b^	7.75 ± 0.4 ^b^	3.25 ± 0.2 ^b^	1.75 ± 0.3 ^abc^	1.25 ± 0.2 ^cd^	1.25 ± 0.5 ^b^
6	5.00 ± 0.4 ^ab^	4.25 ± 0.2 ^a^	4.63 ± 0.5 ^a^	2.13 ± 0.1 ^b^	11.00 ± 0.4 ^a^	7.00 ± 0 ^b^	4.00 ± 0.2 ^ab^	1.38 ± 0.2 ^bc^	1.00 ± 0 ^cd^	0.25 ± 0.3 ^b^
7	5.13 ± 0.3 ^ab^	2.63 ± 0.3 ^d^	2.63 ± 0.3 ^bc^	1.50 ± 0.2 ^b^	0.00 ± 0 ^d^ (NA)	7.00 ± 0 ^b^	3.88 ± 0.1 ^ab^	2.13 ± 0.2 ^ab^	0.88 ± 0.1 ^d^	0.13 ± 0.1 ^b^ (NA)
8	5.25 ± 0.4 ^ab^	4.13 ± 0.3 ^ab^	3.63 ± 0.2 ^ab^	1.00 ± 0 ^b^	0.38 ± 0.2 ^d^	8.13 ± 0.4 ^b^	3.88 ± 0.2 ^ab^	2.63 ± 0.3 ^a^	1.13 ± 0.1 ^b^	0.13 ± 0.1 ^b^

NA: not applied, means within each column for each division with common superscript letters are significantly different (*p* ≤ 0.05).

**Table 4 vaccines-11-01638-t004:** Cytokine parameters through IL-1β, CD4, LYZ, and NO in all chicken groups.

Age (Days)		Groups							
		1	2	3	4	5	6	7	8
IL1β (pg/mL)	17	280.2 ± 7.8 ^a^	220 ± 4.9 ^d^	274.2 ± 2 ^ab^	237.8 ± 14 ^cd^	256.6 ± 1 ^abc^	243.2 ± 0.7 ^bcd^	260.8 ± 11.8 ^abc^	263.4 ± 0.2 ^abc^
	21	312.2 ± 15.4 ^bc^	376.8 ± 1.7 ^ab^	349.2 ± 22.8 ^abc^	410.8 ± 0.5 ^a^	256.2 ± 46.5 ^c^	377 ± 3.7 ^ab^	406.2 ± 7.8 ^ab^	374.4 ± 22.3 ^ab^
	28	364.8 ± 14 ^ab^	264.6 ± 5.1 ^c^	323.2 ± 0.7 ^a^	306 ± 2.4 ^ab^	256.4 ± 1.5 ^c^	268.6 ± 9.6 ^bc^	258.2 ± 14.2 ^c^	290.6 ± 9.6 ^abc^
	31	284.4 ± 8.3	319.6 ± 18.1	318.4 ± 20.576	263.8 ± 37.2	299.2 ± 7.8	255.2 ± 4.2	281.4 ± 1	303.8 ± 6.2
CD4 (ng/mL)	17	4.4 ± 0.2 ^b^	5.0 ± 0 ^ab^	5.0 ± 0 ^ab^	4.6 ± 0.2 ^b^	4.4 ± 0.2 ^b^	5.0 ± 0 ^ab^	5.0 ± 0 ^ab^	5.6 ± 0.2 ^a^
	21	5.0 ± 0 ^bc^	6.0 ± 0 ^a^	5.6 ± 0.2 ^ab^	6.0 ± 0 ^a^	4.6 ± 0.2 ^c^	4.6 ± 0.2 ^c^	5.0 ± 0 ^bc^	5.0 ± 0 ^bc^
	28	5.0 ± 0 ^c^	5.0 ± 0 ^c^	5.0 ± 0 ^c^	5.6 ± 0.2 ^bc^	5.2 ± 0.2 ^c^	6.0 ± 0 ^ab^	5.6 ± 0.2 ^bc^	6.4 ± 0.2 ^a^
	31	5.6 ± 0.2 ^b^	5.6 ± 0.2 ^b^	5.0 ± 0 ^b^	5.4 ± 0.2 ^b^	6.0 ± 0 ^ab^	5.0 ± 0 ^b^	6.0 ± 0 ^ab^	7.2 ± 0.7 ^a^
LYZ (ng/mL)	17	112.2 ± 1.7 ^b^	113.8 ± 0.7 ^ab^	116.6 ± 01 ^ab^	115 ± 1.2 ^ab^	115 ± 2.4 ^ab^	111.4 ± 1 ^b^	115.6 ± 0.2 ^ab^	119.2 ± 0.5 ^a^
	21	112.2 ± 0.7 ^cd^	115 ± 0 ^bc^	109.6 ± 1.5 ^d^	113.4 ± 0.2 ^c^	111.6 ± 1 ^cd^	120.6 ± 0.2 ^a^	112.2 ± 0.73 ^d^	118.2 ± 0.5 ^ab^
	28	107.2 ± 1.7 ^c^	109 ± 0 ^bc^	111.4 ± 2.7 ^bc^	113.8 ± 0.5 ^ab^	106.4 ± 0.2 ^c^	109.2 ± 0.5 ^bc^	111.8 ± 0.8 ^bc^	119.8 ± 1.7 ^a^
	31	104.4 ± 1 ^c^	116.4 ± 0.2 ^bc^	113.4 ± 2.7 ^bc^	117.4 ± 0.5 ^bc^	113.2 ± 0.490 ^bc^	119.8 ± 7.8 ^ab^	132.6 ± 3.4 ^a^	123 ± 0 ^ab^
NO (μmol/L)	17	62.2 ± 1.7 ^b^	63.8 ± 0.7 ^ab^	66.6 ± 1 ^ab^	65 ± 1.2 ^ab^	65 ± 2.4 ^ab^	61.4 ± 1 ^b^	65.6 ± 0.2 ^ab^	69.2 ± 0.5 ^a^
	21	62.2 ± 0.7 ^cd^	65 ± 0 ^bc^	59.6 ± 1.5 ^d^	63.4 ± 0.2 ^c^	61.6 ± 1 ^cd^	70.6 ± 0.2 ^a^	62.2 ± 0.7 ^cd^	68.2 ± 0.5 ^ab^
	28	57.2 ± 1.7 ^c^	59 ± 0 ^bc^	61.4 ± 2.7 ^bc^	63.8 ± 0.5 ^ab^	56.8 ± 0.4 ^c^	59.2 ± 0.5 ^bc^	61.8 ± 0.8 ^bc^	69.8 ± 1.7 ^a^
	31	54.4 ± 1 ^c^	66.4 ± 0.2 ^bc^	63.4 ± 2.7 ^bc^	67.4 ± 1 ^bc^	63.2 ± 0.5 ^bc^	69.8 ± 7.8 ^ab^	82.6 ± 3.4 ^a^	73 ± 0 ^ab^

Means within each row for each division with common superscript letters are significantly different (*p* ≤ 0.05).

## Data Availability

The datasets used and/or analyzed during the current study are available from the corresponding author on reasonable request.

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
