# Peer review of "Superior Efficacy of Apathogenic Genotype I (V4) over Lentogenic Genotype II (LaSota) Live Vaccines against Newcastle Disease Virus Genotype VII.1.1 in Pathogen-Associated Molecular Pattern-H9N2 Vaccinated Broiler Chickens"

_vaccines, 2023, doi:10.3390/vaccines11111638_

Round 1
Reviewer 1 Report
-
The high mortality rate after chanllenged with genotype VII 1.1 NDV in LaSota-vaccinated group is
abnormally. Especially the HI titers of ND in LaSota-vaccinated group is 4.13 (Table 3) at 28DO. And the current consensus is that the immunogenicity of V4 is inferior to La Sota. Thus, the certainty of the results is questionable.
The grammar need improvement, additional round of editing by a native speaker will be beneficial to the manuscript.
Author Response
Reviewer 1
- The high mortality rate after chanllenged with genotype VII 1.1 NDV in LaSota-vaccinated group is abnormally. Especially the HI titers of ND in LaSota-vaccinated group is 4.13 (Table 3) at 28DO. And the current consensus is that the immunogenicity of V4 is inferior to La Sota. Thus, the certainty of the results is questionable.
Response:
Thank you sir for your very important question. The answer of this question is explained briefly in the discussion (lines 348-375 and 411-440). The twice live vaccination using LaSota didn’t offer higher protection due to the following:
- The humoral immune response against non–genotype-matched vaccines (GII) does not have the same specificity for F and HN genes, probably resulting in lower levels of virus neutralization and in turn more severe lesions in birds. In addition, the closer the degree of homology between the virus outbreak and used vaccine seed (affecting the level of specific humoral immune antibodies against F gene and HN gene of the challenge virus), the better protection from viremia and the lower virus load in organs while the decrease in the identity of common epitopes between the vaccine and the vNDV field strains may result in expanded protective antibodies (Miller et al., 2007, 2013; Mohamed et al., 2016; Liu et al., 2017; Yang et al., 2017; Shahar et al., 2018). Tabatabaeizadeh [65] demonstrated that the number of identical linear and conformational neutralizing epitopes of F and HN proteins (n=15) of the vaccine strains compared to the VII.2 NDV were the follows: (V4=14) > (I-2=11) > (Ulster, VG/GA, and R2B =10) > (F=9) > (LaSota=7) > (PHY.LMV.42=6) > (B1=5). In the same previous study, V4 and I-2 vaccine strains showed the highest number of identical epitopes compared to the other vaccine strains, while B1, PHY.LMV.42, and LaSota strains revealed the lowest epitope identity compared to VII.2 NDV “especially in F protein”. Under immunological pressure, the estimation of evolutionary distance confirmed that the number of amino acids substitutions per site between sequences was higher and changed more rapidly for HN compared to F protein [66-73].
- The study of Sultan et al. (2022) showed mortality rate of 23% in 28-days-old broiler chickens following intranasal challenge by vND genotype VII.1.1 and vaccination with both live LaSota and inactivated genotype II NDV vaccines although the HI titers at 28 days old (time of challenge) was 4.2 (the same as in our work). The higher mortality rates in LaSota vaccinated chickens in our study might be attributed to the double vaccinations with LaSota vaccine without any inactivated vaccine and the intramuscular route of challenge vs intranasal route used by Sultan et al.
- Recently, Liu et al., 2022 recorded that 4 log2 HI titers (obtained from humoral or maternally derived antibodies) offered only 50% protection against virulent genotype VII NDV strain JSC0804 strain intraocularly and intranasally. (Liu, M.; Shen, X.; Yu, Y.; Li, J.; Fan, J.; Jia, X.; Dai, Y. Effect of Different Levels of Maternally Derived Genotype VII Newcastle Disease Virus-Specific Hemagglutination Inhibition Antibodies on Protection against Virulent Challenge in Chicks. Viruses 2023, 15, 1840. https://doi.org/10.3390/v15091840)
- Regarding the immunogenicity of V4 compared to LaSota, the HI titers at time of challenge in our study (28 days) were higher numerically only but non-significantly differ between V4 and LaSota. The superior protection of V4 could be also attributed to the higher local immunity induced in the intestinal tract (enterotropic strain) than LaSota (pneumotropic strain) against velogenic viscerotropic genotype VII.1.1 NDV. Positive correlations between the ND live vaccine distribution and its specific immunoglobulin A (IgA) production were observed. Perozo et al. 2008 [60] recorded higher IgA levels in the intestinal tract, bile and intestine for the VG/GA (enterotropic, a pathogenic) strain than LaSota (pneumotropic, lentogenic) which induced higher levels in the trachea. The replication pattern of VG/GA strain induced a stronger localized mucosal immune response in the shown by an increased production of NDV-specific IgA. This feature may represent a competitive advantage after challenge with velogenic viscerotropic NDV owing to the massive destruction of intestinal lymphoid areas and extensive ulceration of overlying intestinal epithelium associated with active replication of this virus [61, Brown et al. 1999]. However, this point should be briefly illustrated in further research work.
Comments on the Quality of English Language
The grammar need improvement, additional round of editing by a native speaker will be beneficial to the manuscript.
- Done. Thanks Sir.
Reviewer 2 Report
The paper titled " Superior efficacy of apathogenic genotype I (V4) over lentogenic genotype II (LaSota) live vaccines against Newcastle disease virus genotype VII.1.1 in pathogen-associated molecular pattern-H9N2 vaccinated broiler chickens" by Ahmed R. Elbestawy et al. analyzed the effectiveness of apathogenic genotype I (V4) and lentogenic genotype II (LaSota) strains of live Newcastle disease virus (NDV) vaccines in combination with a PAMP-H9N2 avian influenza vaccine when challenged with velogenic NDV genotype VII.1.1 (vNDV-VII.1.1). The authors point out that two successive vaccinations of broilers with the live V4 NDV vaccine provided better protection against the vNDV-VII.1.1 challenge compared to the LaSota vaccine. Additionally, combining PAMP-H9N2 with live NDV vaccines induced higher levels of protection than using the live vaccine alone. The work is interesting and suitable for publishing in Vaccines, but I think some revisions need to be introduced before publication.
Major issues:
Figure 1: “DPI” should be “DPC or days post challenge”.
Minor issues:
1. Line 42: “thefore” should be “therefore”.
2. Line 345: “Some pervious Egyptian” should be “Some previous Egyptian”.
3. Line 390: “more protection of chcikens” should be “more protection of chickens”.
4. Line 396: “demomstrated that” should be “demonstrated that”.
Minor editing of English language required.
Author Response
Reviewer 2
Comments and Suggestions for Authors
The paper titled " Superior efficacy of apathogenic genotype I (V4) over lentogenic genotype II (LaSota) live vaccines against Newcastle disease virus genotype VII.1.1 in pathogen-associated molecular pattern-H9N2 vaccinated broiler chickens" by Ahmed R. Elbestawy et al. analyzed the effectiveness of apathogenic genotype I (V4) and lentogenic genotype II (LaSota) strains of live Newcastle disease virus (NDV) vaccines in combination with a PAMP-H9N2 avian influenza vaccine when challenged with velogenic NDV genotype VII.1.1 (vNDV-VII.1.1). The authors point out that two successive vaccinations of broilers with the live V4 NDV vaccine provided better protection against the vNDV-VII.1.1 challenge compared to the LaSota vaccine. Additionally, combining PAMP-H9N2 with live NDV vaccines induced higher levels of protection than using the live vaccine alone. The work is interesting and suitable for publishing in Vaccines, but I think some revisions need to be introduced before publication.
Major issues:
Figure 1: “DPI” should be “DPC or days post challenge”.
- Thank you so much for your revision sir. The abbreviation is corrected to DPC. Line 199.
Minor issues:
- Line 42: “thefore” should be “therefore”.
- Line 42.
- Line 345: “Some pervious Egyptian” should be “Some previous Egyptian”.
- Line 369.
- Line 390: “more protection of chcikens” should be “more protection of chickens”.
- Line 427.
- Line 396: “demomstrated that” should be “demonstrated that”.
- Line 433.
Comments on the Quality of English Language
Minor editing of English language required.
- Thanks Sir.
Reviewer 3 Report
This study provide an efficient way for clinical prevention of ND. The experiment design is good, however, there are some concerns as follow:
1. Line 31, "46.66% in G6", there are no details about the mortality, please show the birds numbers and all treatments during the whole experiment in a table.
2. Line 84, "subtype H9N2 prepared by PAMP technology (PAMP-H9N2)", is there any information about this vaccine?
3. Line 142, -80oC should be -80℃
4. Line 154-160, why the serum was collected at 17, 21, 28, and 31 DO for cytokines detection, but 4 spleen samples were collected for examination?
5. Table 2, The Score scale in Line 135 was 1, 2, 3 and 4, why the standard deviation had three valid numbers?There should be only one valid number.
6. Figure 7, it should be shown in two figures, A and B
7. Table 3, maybe only one valid number is good for understand
8. Table 4, the valid numbers are not identical.
9. Line 308, "IL-4 and IL-10 IF" should be "IL-4, IL-10 and IF-γ"
10. It seems twice ND vaccination has no obvious effect on the HI titers in Table 3, please discuss the reason
Author Response
Reviewer 3
Comments and Suggestions for Authors
This study provide an efficient way for clinical prevention of ND. The experiment design is good, however, there are some concerns as follow:
Responses:
- Line 31, "46.66% in G6", there are no details about the mortality, please show the birds numbers and all treatments during the whole experiment in a table.
- Thank you sir for your important notice and comment. We added a supplementary table S1 (line 192).
Actually, 2 birds were removed on day 0, 3, 7, 10 and 14 days for histopathology. The histopathology of 10 and 14 days was the same, so we missed to add 14 days in the first submitted version. But we added it now.
Mortality rates in all groups (each group contained 15 birds after exclusion of 10 birds used for histopathology)
Groups (15 birds each) |
Days Post Challenge |
Total Mortality |
|||||||||
1 |
2 |
3 |
4 |
5 |
6 |
7 |
8 |
9 |
10 |
||
1 |
0 |
0 |
0 |
0 |
0 |
0 |
0 |
0 |
0 |
0 |
0 |
2 |
0 |
0 |
1 |
1 |
1 |
2 |
1 |
0 |
0 |
0 |
6 (40%) |
3 |
0 |
2 |
2 |
5 |
4 |
1 |
1 |
NA |
NA |
NA |
15 (100%) |
4 |
0 |
0 |
0 |
0 |
0 |
0 |
0 |
0 |
0 |
0 |
0 (0%) |
5 |
0 |
0 |
0 |
0 |
0 |
0 |
0 |
0 |
0 |
0 |
0 (0%) |
6 |
0 |
0 |
1 |
2 |
4 |
0 |
0 |
0 |
0 |
0 |
7 (46.66%) |
7 |
0 |
3 |
5 |
5 |
2 |
NA |
NA |
NA |
NA |
NA |
15 (100%) |
8 |
0 |
0 |
0 |
0 |
0 |
0 |
0 |
0 |
0 |
0 |
0 (0%) |
NA: not applied as all birds were died.
- Line 84, "subtype H9N2 prepared by PAMP technology (PAMP-H9N2)", is there any information about this vaccine?
- Added in lines 77-82.
- Line 142, -80oC should be -80℃
- Line 152.
- Line 154-160, why the serum was collected at 17, 21, 28, and 31 DO for cytokines detection, but 4 spleen samples were collected for examination?
- The sera for detection of some immune mediator levels using enzyme linked immuno-sorbent assay (ELISA) kits which were based on the principle of double antibody sandwich technology including interleukin-1b (IL-1b), cluster of differentiation 4 (CD4), lysozyme (LYZ), and nitric oxide (NO) using enzyme linked immuno-sorbent assay (ELISA) kits (Novatein Bio, Massachusetts, USA). While, 4 spleen samples were collected from each chicken group at 31 DO and their homogenates were examined for gene expression of IL-4, IL-10, and interferon-g (IF-g).
- Table 2, The Score scale in Line 135 was 1, 2, 3 and 4, why the standard deviation had three valid numbers?There should be only one valid number.
- Line 285.
- Figure 7, it should be shown in two figures, A and B
- Line 295.
- Table 3, maybe only one valid number is good for understand
- Line 304.
- Table 4, the valid numbers are not identical.
- Line 317.
- Line 308, "IL-4 and IL-10 IF" should be "IL-4, IL-10 and IF-γ"
- Line 327.
- It seems twice ND vaccination has no obvious effect on the HI titers in Table 3, please discuss the reason
Response: discussed in lines 348-375 and 411-440
- The humoral immune response against non–genotype-matched vaccines (GII) does not have the same specificity for F and HN genes, probably resulting in lower levels of virus neutralization and in turn more severe lesions in birds. In addition, the closer the degree of homology between the virus outbreak and used vaccine seed (affecting the level of specific humoral immune antibodies against F gene and HN gene of the challenge virus), the better protection from viremia and the lower virus load in organs while the decrease in the identity of common epitopes between the vaccine and the vNDV field strains may result in expanded protective antibodies (Miller et al., 2007, 2013; Mohamed et al., 2016; Liu et al., 2017; Yang et al., 2017; Shahar et al., 2018). Tabatabaeizadeh [65] demonstrated that the number of identical linear and conformational neutralizing epitopes of F and HN proteins (n=15) of the vaccine strains compared to the VII.2 NDV were the follows: (V4=14) > (I-2=11) > (Ulster, VG/GA, and R2B =10) > (F=9) > (LaSota=7) > (PHY.LMV.42=6) > (B1=5). In the same previous study, V4 and I-2 vaccine strains showed the highest number of identical epitopes compared to the other vaccine strains, while B1, PHY.LMV.42, and LaSota strains revealed the lowest epitope identity compared to VII.2 NDV “especially in F protein”. Under immunological pressure, the estimation of evolutionary distance confirmed that the number of amino acids substitutions per site between sequences was higher and changed more rapidly for HN compared to F protein [66-73].
- The study of Sultan et al. (2022) showed mortality rate of 23% in 28-days-old broiler chickens following intranasal challenge by vND genotype VII.1.1 and vaccination with both live LaSota and inactivated genotype II NDV vaccines although the HI titers at 28 days old (time of challenge) was 4.2 (the same as in our work). The higher mortality rates in LaSota vaccinated chickens in our study might be attributed to the double vaccinations with LaSota vaccine without any inactivated vaccine and the intramuscular route of challenge vs intranasal route used by Sultan et al.
- Recently, Liu et al., 2022 recorded that 4 log2 HI titers (obtained from humoral or maternally derived antibodies) offered only 50% protection against virulent genotype VII NDV strain JSC0804 strain intraocularly and intranasally.
(Liu, M.; Shen, X.; Yu, Y.; Li, J.; Fan, J.; Jia, X.; Dai, Y. Effect of Different Levels of Maternally Derived Genotype VII Newcastle Disease Virus-Specific Hemagglutination Inhibition Antibodies on Protection against Virulent Challenge in Chicks. Viruses 2023, 15, 1840. https://doi.org/10.3390/v15091840).
- Regarding the immunogenicity of V4 compared to LaSota, the HI titers at time of challenge in our study (28 days) were higher numerically only but non-significantly differ between V4 and LaSota. The superior protection of V4 could be also attributed to the higher local immunity induced in the intestinal tract (enterotropic strain) than LaSota (pneumotropic strain) against velogenic viscerotropic genotype VII.1.1 NDV. Positive correlations between the ND live vaccine distribution and its specific immunoglobulin A (IgA) production were observed. Perozo et al. 2008 [60] recorded higher IgA levels in the intestinal tract, bile and intestine for the VG/GA (enterotropic, a pathogenic) strain than LaSota (pneumotropic, lentogenic) which induced higher levels in the trachea. The replication pattern of VG/GA strain induced a stronger localized mucosal immune response in the shown by an increased production of NDV-specific IgA. This feature may represent a competitive advantage after challenge with velogenic viscerotropic NDV owing to the massive destruction of intestinal lymphoid areas and extensive ulceration of overlying intestinal epithelium associated with active replication of this virus [61, Brown et al. 1999]. However, this point should be briefly illustrated in further research work.
Reviewer 4 Report
The authors describe a study comparing two different ND-vaccines representing two different genotype and include a commercial inactivated AIV H9 vaccine. The study was done in commercial broilers, with maternal derived antibodies against both, AIV-H9 and NDV specific antibodies.
They present data that after challenge infection with a virulent NDV genotype VII.1.1 virus, only the V1 vaccinated groups, regardless of their AIV H9 immune status were clinically protected.
Beside data on mortality, they present data on serological response in weekly intervals, shedding data obtained by RT-qPCR, histology and cytokine parameters.
The authors claim, that differences in clinical outcome, is due to antigenic differences. However, they did not present specific data on that. This would need serological testing of obtained sera against all three genotypes involved, i.e. V1 (genotype I) LaSota (gentyope II) and challenge virus (genotype VI.1.1). Besides, other factors influencing vaccine efficacy like maternal derived antibodies, vaccine take, underlying immune suppressive factors have to be addressed and excluded.
In addition, in order to evaluate quality of the results, specifics on correlation of serological response and clinical outcome are needed.
Besides, certain uncertainties have to be clarified.
What was the basis for the sample size estimation (n=3 for viral shedding, n= 8 for serology).
Were always the same animals selected for sampling ?
How many animals were positive?
Please address why animals of LaSota vaccinated animal (group G2 and G6) differ on day 28.
Please address the question why AIV H9 antibody increase 7 days after NDV challenge infection.
What was considered 100% for the group: 25 animals total infected, but two animals removed on day 0, 3, 7 and 10. More precise showing be the number of dead animals.
Did all chicken die spontaneous or were chicken euthanized.
Histological data should be presented in a time dependent manner, i.e. time after infection has to be considered.
In addition more data on clinical course should be given, for example clinical scoring over time, like in an ICPI experiment.
Altogether a more a more balanced presentation is needed. For example it is well known, that single epitopes within the outer glycoproteins vary, by the effect of polyclonal response relatively small. Those results have to be considered.
no comments
Author Response
Reviewer 4
Comments and Suggestions for Authors
The authors describe a study comparing two different ND-vaccines representing two different genotype and include a commercial inactivated AIV H9 vaccine. The study was done in commercial broilers, with maternal derived antibodies against both, AIV-H9 and NDV specific antibodies.They present data that after challenge infection with a virulent NDV genotype VII.1.1 virus, only the V1 vaccinated groups, regardless of their AIV H9 immune status were clinically protected. Beside data on mortality, they present data on serological response in weekly intervals, shedding data obtained by RT-qPCR, histology and cytokine parameters. The authors claim, that differences in clinical outcome, is due to antigenic differences. However, they did not present specific data on that. This would need serological testing of obtained sera against all three genotypes involved, i.e. V1 (genotype I) LaSota (gentyope II) and challenge virus (genotype VI.1.1). Besides, other factors influencing vaccine efficacy like maternal derived antibodies, vaccine take, underlying immune suppressive factors have to be addressed and excluded.
- In addition, in order to evaluate quality of the results, specifics on correlation of serological response and clinical outcome are needed. Besides, certain uncertainties have to be clarified.
Response: Thank you sir for your notice. The correlation of serological response and clinical outcome are all included in discussion (lines 352-358; 374-375 and 385-410). Also, all your uncertainties are replied briefly in the following comments.
- What was the basis for the sample size estimation (n=3 for viral shedding, n= 8 for serology).
Response: Thank you sir for your comment. These were the minimum representative numbers for each group from our point of view. Unfortunately, we were not able to test more than these samples as this was a non-funded research.
- Were always the same animals selected for sampling ?
Response: Yes, birds used for sampling were signed. However, if they were died like in G7, 3, 6 and 2, other birds toke place.
- How many animals were positive?
Response: Discussed briefly in table S2.
Supplementary table S2: Clinical disease score in all groups
Chicken group |
Clinical signs |
Days after inoculation Number of chickens with specific signs |
Total score |
|||||||||||
1 |
2 |
3 |
4 |
5 |
6 |
7 |
8 |
9 |
10 |
|||||
G1 |
Normal |
15 |
15 |
15 |
12 |
10 |
15 |
15 |
15 |
15 |
15 |
142 X 0 = 0 |
||
Sick |
0 |
0 |
0 |
3 |
5 |
0 |
0 |
0 |
0 |
0 |
8 X 1 = 8 |
|||
Dead |
0 |
0 |
0 |
0 |
0 |
0 |
0 |
0 |
0 |
0 |
0 X 2 = 0 |
|||
Total score = 8/150 = 0.1d |
||||||||||||||
G2 |
Normal |
0 |
0 |
12 |
10 |
7 |
5 |
5 |
5 |
6 |
8 |
58 X 0 = 0 |
||
Sick |
0 |
0 |
2 |
3 |
5 |
5 |
4 |
4 |
3 |
1 |
27 X 1 = 27 |
|||
Dead |
0 |
0 |
1 |
2 |
3 |
5 |
6 |
6 |
6 |
6 |
35 X 2 = 70 |
|||
Total score = 97/150 = 0.6c |
||||||||||||||
G3 |
Normal |
15 |
6 |
2 |
0 |
0 |
0 |
0 |
0 |
0 |
0 |
23 X 0 = 0 |
||
Sick |
0 |
7 |
9 |
6 |
2 |
1 |
0 |
0 |
0 |
0 |
25 X 1 = 25 |
|||
Dead |
0 |
2 |
4 |
9 |
13 |
14 |
15 |
15 |
15 |
15 |
102 X 2 = 204 |
|||
Total score = 229/150 = 1.5a |
||||||||||||||
G4 |
Normal |
15 |
15 |
15 |
15 |
15 |
15 |
15 |
15 |
15 |
15 |
150 X 0 = 0 |
||
Sick |
0 |
0 |
0 |
0 |
0 |
0 |
0 |
0 |
0 |
0 |
0 X 1 = 0 |
|||
Dead |
0 |
0 |
0 |
0 |
0 |
0 |
0 |
0 |
0 |
0 |
0 X 2 = 0 |
|||
Total score = 0/150 = 0d |
||||||||||||||
G5 |
Normal |
15 |
15 |
13 |
10 |
12 |
14 |
15 |
15 |
15 |
15 |
139 X 0 = 0 |
||
Sick |
0 |
0 |
2 |
5 |
3 |
1 |
0 |
0 |
0 |
0 |
11 X 1 = 11 |
|||
Dead |
0 |
0 |
0 |
0 |
0 |
0 |
0 |
0 |
0 |
0 |
0 X 2 = 0 |
|||
Total score = 11/150 = 0.1d |
||||||||||||||
G6 |
Normal |
0 |
0 |
11 |
8 |
3 |
2 |
3 |
3 |
4 |
6 |
40 X 0 = 0 |
||
Sick |
0 |
0 |
3 |
4 |
5 |
6 |
5 |
5 |
4 |
2 |
34 X 1 = 34 |
|||
Dead |
0 |
0 |
1 |
3 |
7 |
7 |
7 |
7 |
7 |
7 |
46 X 2 = 92 |
|||
Total score = 126/150 = 0.8b |
||||||||||||||
G7 |
Normal |
15 |
3 |
0 |
0 |
0 |
0 |
0 |
0 |
0 |
0 |
18 X 0 = 0 |
||
Sick |
0 |
9 |
7 |
2 |
0 |
0 |
0 |
0 |
0 |
0 |
18 X 1 = 18 |
|||
Dead |
0 |
3 |
8 |
13 |
15 |
15 |
15 |
15 |
15 |
15 |
114 X 2 = 228 |
|||
Total score = 246/150 = 1.6a |
||||||||||||||
G8 |
Normal |
15 |
15 |
15 |
15 |
15 |
15 |
15 |
15 |
15 |
15 |
150 X 0 = 0 |
||
Sick |
0 |
0 |
0 |
0 |
0 |
0 |
0 |
0 |
0 |
0 |
0 X 1 = 0 |
|||
Dead |
0 |
0 |
0 |
0 |
0 |
0 |
0 |
0 |
0 |
0 |
0 X 2 = 0 |
|||
Total score = 0/150 = 0d |
||||||||||||||
Clinical signs included general signs of depression, and decreased feed intake; respiratory signs such as sneezing, rales, nasal and ocular discharge, conjunctivitis, coughing, head swelling; enteric signs such as greenish diarrhea; nervous signs such as head shaking, torticollis, and lateral recumbency. While, post mortem lesions of tracheitis, pneumonia, proventricular hemorrhages, enteritis, petechial hemorrhages on ileocecal tonsils, splenitis, hepatic congestion with distended gall bladder, and nephritis.
- Please address why animals of LaSota vaccinated animal (group G2 and G6) differ on day 28.
Response: The interference with specific maternal immunity to genotype II through delayed the serological response of LaSota vaccination for more than 28 days of age in G6 (2.13 log2). It is well known that specific maternal antibodies could delay the recognition of live vaccination; however, in G2 the use of inactivated vaccine PAMP-H9N2 could compensate this point through the faster recognition of PAMP by PRRs inducing higher immune enhancement (especially innate and cellular immunity) which may predispose the higher humoral immunity appeared at 28 days old (4.13 log2).
- Please address the question why AIV H9 antibody increase 7 days after NDV challenge infection.
Response: There is no any correlation between AIV H9 antibodies and NDV challenge. The titers of LPAI-H9N2 increased at 35 days in the vaccinated groups 1, 2 and 4 in comparison to 28 days in the same groups and also in comparison to the other non-H9N2-vaccinated groups due to the induced humoral immunity of the H9N2 vaccine itself. The delay in this immune response up to more than 28 days (35 days actually in this study) may be attributed to the interference with maternal immunity (Pan et al., 2022).
- Random swabs were tested for H9N2 pre- and post-challenge to be sure of negative H9N2 infection. (Pan, X., Su, X., Ding, P. et al. Maternal-derived antibodies hinder the antibody response to H9N2 AIV inactivated vaccine in the field. Animal Diseases 2, 9 (2022). https://doi.org/10.1186/s44149-022-00040-0)
- What was considered 100% for the group: 25 animals total infected, but two animals removed on day 0, 3, 7 and 10. More precise showing be the number of dead animals.
Response: Actually, 2 birds were removed on day 0, 3, 7, 10 and 14 days for histopathology. The histopathology of 10 and 14 days was the same, so we missed to add 14 days in the first submitted version. But we added it now.
Mortality rates in all groups (each group contained 15 birds after exclusion of 10 birds used for histopathology)
Groups (15 birds each) |
Days Post Challenge |
Total Mortality |
|||||||||
1 |
2 |
3 |
4 |
5 |
6 |
7 |
8 |
9 |
10 |
||
1 |
0 |
0 |
0 |
0 |
0 |
0 |
0 |
0 |
0 |
0 |
0 |
2 |
0 |
0 |
1 |
1 |
1 |
2 |
1 |
0 |
0 |
0 |
6 (40%) |
3 |
0 |
2 |
2 |
5 |
4 |
1 |
1 |
NA |
NA |
NA |
15 (100%) |
4 |
0 |
0 |
0 |
0 |
0 |
0 |
0 |
0 |
0 |
0 |
0 (0%) |
5 |
0 |
0 |
0 |
0 |
0 |
0 |
0 |
0 |
0 |
0 |
0 (0%) |
6 |
0 |
0 |
1 |
2 |
4 |
0 |
0 |
0 |
0 |
0 |
7 (46.66%) |
7 |
0 |
3 |
5 |
5 |
2 |
NA |
NA |
NA |
NA |
NA |
15 (100%) |
8 |
0 |
0 |
0 |
0 |
0 |
0 |
0 |
0 |
0 |
0 |
0 (0%) |
NA: not applied as all birds were died.
- Did all chicken die spontaneous or were chicken euthanized.
Response: The examined chickens for histopathology were euthanized.
- Histological data should be presented in a time dependent manner, i.e. time after infection has to be considered.
Response: Sorry for this mistake Sir, time dependent manner post infection is now included in the results.
- In addition more data on clinical course should be given, for example clinical scoring over time, like in an ICPI experiment.
Response: Thank you so much sir for your constructive comment. Tables (S1, S2) for clinical disease score has been indicated. We totally agree with you sir that this score will clarify, and add much value to the obtained results.
- Altogether a more a more balanced presentation is needed. For example it is well known, that single epitopes within the outer glycoproteins vary, by the effect of polyclonal response relatively small. Those results have to be considered.
Response: Despite Newcastle disease is one serotype, several genotypes of NDV were recorded. NDV strains are divided into two major classes, namely class I and class II. Class I comprises non-virulent strains and consists of one genotype and 3 sub-genotypes. However, class II includes virulent and non-virulent strains and consists of at least 20 genotypes (I–XXI) and multiple sub-genotypes following the exclusion of genotype XV which contains recombinant sequences [Dimitrov et al., 2019]. The outer surface proteins such as hemagglutinin-neuraminidase (HN) and F which are located on the NDV envelope are important for viral virulence, tropism, and immunoprotection through neutralizing antibodies. The antigenic differences (neutralizing epitopes) between the circulating virulent viruses and the vaccine strains may be responsible for the inadequate vaccine efficacy [Liu et al., 2017, Dimitrov et al., 2019, Miller et al., 2007, Dortmans et al., 2012]. Besides, the antigenic similarity between the circulating velogenic (vNDV) and the vaccine strain could improve the induced vaccinal protection in terms of reduced infectivity and virus shedding [Miller et al., 2009, 2013, Yang et al., 2017, Shahar et al., 2018].
- Comments on the Quality of English Language
no comments.
Response: Thank you sir.
Round 2
Reviewer 1 Report
This manuscript has been revised according to the comments and could be considered for publication.
Author Response
Thank you
Reviewer 3 Report
All point had been answered.
Author Response
Thank you
Reviewer 4 Report
The authors did not respond to the major concern: data on antigenic differences were not included/generated.
english laguage seems appropriate
Author Response
Reviewer 4
Thank you sir for your comments and I hope that all responses to your important questions in the first round have been done correctly point by point.
Comments and Suggestions for Authors (second round)
- The authors did not respond to the major concern: data on antigenic differences were not included/generated.
Response: Thank you sir for your constructive comment. We included a paragraph in lines 336-349 that discusses the previously reported data on antigenic differences to reply to this issue. Please note that our study was planned to confirm the theory of higher genetic and antigenic matching within F and HN proteins is essential for higher protection against vNDV genotype VII.1.1 in broiler chickens and we chose to compare genotype I (V4) or genotype II (LaSota) strains of live NDV vaccines especially in the presence of PAMP-H9N2 vaccination as an enhancer for innate immune response. Also, the paragraph in lines 436-363 discusses more research about the antigenic relatedness between vNDV-VII.1.1 and different ND vaccine strains.
Lines 336-352: Although NDV is one serotype, several genotypes were recorded. NDV strains are divided into two major classes, namely class I and class II. Class I comprises non-virulent strains and consists of one genotype and 3 sub-genotypes. However, class II includes virulent and non-virulent strains and consists of at least 20 genotypes (I–XXI) and multiple sub-genotypes following the exclusion of genotype XV which contains recombinant sequences [4]. The outer surface proteins such as hemagglutinin-neuraminidase (HN) and F which are located on the NDV envelope are important for viral virulence, tropism, and immune protection through neutralizing antibodies. The antigenic differences in neutralizing epitopes between the circulating virulent viruses (e.g. genotype VII.1.1) and the vaccine strains may be responsible for the inadequate vaccine efficacy [4, 13, 15, 16]. Furthermore, the higher matching percent of aa in the F and HN proteins of NDV vaccines and the field virus is essential for enhancing the vaccine protection in terms of reduced infectivity and minimized viral shedding rate [6,14,15,17-19, 46-48]. Previously, high protection and reduction of viral shedding have been reported in birds vaccinated with recombinant vaccines containing genotype I (D26) and challenged by heterologous genotypes VII.1.1, VII.2, and V of NDV [49-52].
Lines 436-463: The main question of this study is how could V4 (genotype-I) vaccine of NDV could provoke higher protection of chickens than LaSota did. To answer this question, the role of both F and HN proteins in the induction of neutralizing antibodies against vNDV should be considered. The HN protein prevents the viral attachment step (anti-HN antibodies), while the F protein inhibits virus spread to the nearby cells (anti-F antibodies), even when the virus enters the cell and multiplies within it [62-64]. Further, Kim et al. [12] reported that the F protein plays the most important role in the protection of virus infection following the use of genotype-matched vaccines. Tabatabaeizadeh [65] recorded that, the number of identical linear and conformational neutralizing epitopes of F and HN proteins (n=15) of the vaccine strains compared to the VII.2 NDV were the follows: (V4=14) > (I-2=11) > (Ulster, VG/GA, and R2B =10) > (F=9) > (LaSota=7) > (PHY.LMV.42=6) > (B1=5). In the same previous study, V4 and I-2 vaccine strains showed the highest number of identical epitopes compared to the other vaccine strains, while B1, PHY.LMV.42, and LaSota strains revealed the lowest epitope identity compared to VII.2 NDV “especially in F protein”. Similar results of aa differences between LaSota and VG/GA were recorded earlier by [60]. Under immunological pressure, the estimation of evolutionary distance confirmed that the number of amino acid substitutions per site between sequences was higher and changed more rapidly for HN compared to F protein [66-73]. Additionally, the decrease in the identity of common epitopes between the vaccine and the vNDV field strains may result in expanded and non-matched protective antibodies, lower levels of virus neutralization, higher rate of viremia, higher severity of the disease, and prolonged virus load in organs [6, 15, 17-19]. In the recent study of Liu et al. [74], the results indicated that the amino acids similarity of LaSota strain with vNDV genotype VII strain was 88.4% for the F protein and 88.5% for the HN protein and that with vNDV genotype IX strain was 92.2% for the F protein and 91.1% for the HN protein. Moreover, the genotype IX strain is genetically closer to LaSota strain than the genotype VII strain and consequently, this higher antigenic similarity can provide better protection against the development of clinical disease and viral shedding.
- Comments on the Quality of English Language: English language seems appropriate.
Response: Thank you so much, sir.
